# Geospatial distribution of Hepatitis E seroprevalence in Nepal, 2021

**Chulwoo Rhee** [1]☺*, **Amy Dighe**[2]☺, **Nishan Katuwal**[3,4], **Haeun Cho**[1], **Ramzi Mraidi**[1], **Dipesh Tamrakar**[3,4,5], **Jacqueline KyungAh Lim**[1], **Nimesh Poudyal**[1], **Il-Yeon Park**[1], **Deok Ryun Kim**[1], **Ritu Amatya**[6], **Rajeev Shrestha**[3,4,7], **Andrew S. Azman**[2,8,9], **Julia Lynch**[1]

1 International Vaccine Institute, Seoul, Republic of Korea, 2 Department of Epidemiology, Johns Hopkins Bloomberg School of Public Health, Baltimore, Maryland, United States of America, 3 Center for Infectious Disease Research and Surveillance, Dhulikhel Hospital, Kathmandu University Hospital, Dhulikhel, Nepal, 4 Research and Development Division, Dhulikhel Hospital, Kathmandu University Hospital, Dhulikhel, Nepal, 5 Department of Community Medicine, Kathmandu University School of Medical Sciences, Dhulikhel, Nepal, 6 Department of Microbiology, Nepal Medical College Teaching Hospital, Kathmandu Nepal, 7 Department of Pharmacology, Kathmandu University School of Medical Sciences, Dhulikhel, Nepal, 8 Geneva Centre for Emerging Viral Diseases, Geneva University Hospitals, Geneva, Switzerland, 9 Division of Tropical and Humanitarian Medicine, Geneva University Hospitals, Geneva, Switzerland

☺ These authors contributed equally to this work.
* rhee275@gmail.com

**Data Availability Statement:** The data and code used in these analyses are available at https://github.com/HopkinsIDD/nepal_hev.

**Funding:** This study was funded by the Bill & Melinda Gates Foundation (INV-039469 to JAL).

## Abstract

### Background

Hepatitis E virus (HEV) causes acute jaundice and poses an important public health problem in low- and middle-income countries. Limited surveillance capacity and suboptimal access to diagnostics leads to under-reporting of HEV infections in affected countries, including Nepal. Serum antibodies against HEV are indicative of past infection. We analyzed existing samples from a nationally representative serosurvey to describe the geospatial distribution and factors associated with HEV seroprevalence in Nepal, as a proxy for infection.

### Methodology/Principle findings

A nationally representative cross-sectional serosurvey of 3,922 individuals ≥2 years old from 975 households spread across 65 wards throughout Nepal was conducted between November 2021 and January 2022. Bio-banked samples were tested for anti-HEV IgG. Seroprevalence and its 95% confidence interval were estimated by age, sex, ecological region, municipality type, and other waterborne-disease related risk factors. Bayesian geostatistical models were fitted to observed seroprevalence data and used to generate high-resolution maps of seroprevalence across Nepal. Available samples from 3,707 participants were tested for anti-HEV IgG, and 3,703 were used for final analysis. We found 20.8% (95% CI: 19.5–22.2) of participants had evidence of prior HEV infection. HEV seroprevalence increased with age, and was higher in males (23.5%, 95% CI: 21.5–25.5) than in females (18.6%, 95% CI: 16.9–20.3). Seroprevalence in hilly (28.9%, 95% CI: 26.6–31.2) and mountain (24.6%, 95% CI: 18.8–30.5) regions were significantly higher than in terai (14.2%, 95% CI: 12.7–15.8). While there was no significant difference between urban and rural populations, the predicted seroprevalence was highest in Kathmandu, the capital of Nepal,

The original cross-sectional field survey was funded by the Global Disease Eradication Fund from the government of Republic of Korea (2020-2025 to JAL). The funders had no role in study design, data collection and analysis, decision to publish, or preparation of the manuscript.

**Competing interests:** The authors have declared that no competing interests exist.

reaching seroprevalence of 50% in some selected area. No statistically significant differences were found for wealth quintile, water source, and toilet facility.

## Conclusions

This study provides population-based serologic evidence that HEV is endemic in Nepal, with the greatest risk of infection in Kathmandu.

### Author summary

Hepatitis E virus spreads through fecal to oral transmission, primarily via contaminated water. The virus can impair liver function and causes frequent outbreaks of acute jaundice in Nepal. Diagnostic tests are often not accessible, meaning many cases of hepatitis E go undetected and it is difficult to know how many people are affected or where to target interventions to protect those most at risk. However, infections can be detected retrospectively by looking for long-lasting antibodies produced by the body in response to the virus. We looked for these antibodies in blood specimens collected from a population representative sample of individuals across Nepal and found that, overall, around 1 in 5 people had likely been infected previously. Risk of past infection varied across the three ecological regions of Nepal and was higher in Hilly and Mountain regions than in the southern Terai. We fitted a geospatial model to our data to map the predicted risk across the country and found it to be highest in Kathmandu, with up to half the population previously infected. This research demonstrates the scale of hepatitis E virus infections in Nepal and identified parts of the Hilly region–particularly Kathmandu–as areas where people are most at risk.

## Introduction

Hepatitis E virus (HEV) causes acute jaundice by impacting liver function. HEV poses a significant public health problem in many low- and middle-income countries (LMICs) where genotypes 1 and 2 are endemic, spreading from person-to-person via contaminated water [1]. Despite causing large outbreaks of acute jaundice and being associated with a high case fatality rate among pregnant women [2], HEV is often considered a neglected disease as the majority of HEV infections remain undetected due to self-limiting clinical characteristics, poor surveillance, and limited access to healthcare [3]. Thus, global burden estimates are highly variable. A systematic review conducted in 2020 estimated approximately 939 million people, corresponding to 1 in 8 individuals, have experienced HEV infection globally and 15–110 million individuals have recent or ongoing HEV infection [4]. Due to limited surveillance capacity and suboptimal access to diagnostics, the true burden of HEV infections in countries like Nepal is likely underreported.

HEV is considered to be endemic in Nepal and was reported to be responsible for about 50% of acute hepatitis occurring in Kathmandu, highlighting the virus's significant impact in urban areas [5]. To date, five major HEV epidemics have been documented, largely affecting urban areas of Kathmandu valley [6]. The earliest documented outbreak was in 1982, though the virus was likely present in the region earlier [7]. The first major outbreak outside of the Kathmandu valley was documented in 2014 and was estimated to affect approximately 7,000 people in Biratnagar city following consumption of contaminated municipal water [8]. Isolates

collected during the outbreak investigation have been classified as HEV genotype 1. It is expected that with continued rapid urbanization in Nepal the population at risk of HEV infection and outbreak may grow.

Despite several studies reporting endemicity of HEV in Nepal, nationally representative disease burden data on HEV are lacking. Previous studies on HEV in Nepal have primarily focused on outbreak investigations and small-scale surveys, leaving a gap in comprehensive national-level data. Population-based HEV epidemiologic data are needed to support informed decision making on strengthening the national waterborne disease surveillance system and to generate a comprehensive control plan. In the absence of consistent case-based surveillance data, seroprevalence can provide an important proxy measure of infection. Therefore, we conducted a study to estimate HEV seroprevalence and describe the geospatial distribution using banked samples from a previously conducted cross-sectional serosurvey for estimating burden of cholera in Nepal.

## Methods

### Ethics statement

The protocol for the original cholera cross-sectional serosurvey was approved by the National Health Research Council (NHRC) of Nepal (NHRC registration number 166/2021), the International Vaccine Institute (IVI) Institutional Review Board (IVI IRB 2020–016). Formal written consent/assent was obtained from study participants before data and blood sample collection began. In addition, separate formal written consent/assent was obtained from study participants for long-term storage and future testing for infectious etiologies. For child participants, formal written consent was obtained from the parent or guardian.

### Survey design

We analyzed questionnaire data and blood samples from a nationally representative cross-sectional serosurvey that was conducted between November 2021 to January 2022 with the original aim of estimating cholera seroincidence across Nepal. Nepal is divided into seven provinces: Koshi, Madhesh, Bagmati, Gandaki, Lumbini, Karnali, and Sudurpashchim. Provinces are divided into districts, districts into municipalities, and municipalities into wards. Each ward can be categorized as either urban or rural depending on its administrative municipality type. The number of households in rural wards typically range between 150–300, while wards in urban area can go up to 800 or more. Apart from administrative classification, Nepal has three distinctive ecological regions (terai, hilly, or mountain). To ensure national representativeness, two-step cluster random sampling was performed. First, 65 wards (primary sampling unit) were selected based on population size and geographic diversity to ensure representativeness across different ecological regions and municipality types. Then 15 households were selected per ward using systematic random sampling after household enumeration, taking into account factors such as household density and accessibility.

All members of selected households, aged ≥2 years who agreed to participate, were interviewed and blood specimens were collected (up to 8mL for adult, 5mL for children). Collected blood samples were processed, and sera were aliquoted up to 4 separate vials, each containing at least 0.5mL, by the trained laboratorians in the field. All aliquoted vials were kept and transported at -20°C to the central laboratory located at the Nepal Medical College. Upon arrival, samples were kept at -80°C freezer until laboratory testing. As part of the consenting, participants were separately consented for long-term bio-banking of collected samples which can be used for other infectious etiologies, except HIV. A household-level questionnaire was completed by the head of each selected household, and individual-level questionnaires were

completed by all participating members. Primary caregiver completed the questionnaires in case toddler or young child was unable to provide responses. Up to three attempts were made to enroll every eligible household member.

## Laboratory analysis

Banked serum vials, one vial from every participant who consented to long-term storage and future analysis, were transported to the Dhulikhel Hospital at -80°C for laboratory analysis. Samples were tested for anti-HEV Immunoglobulin G (IgG) using the commercially available Wantai immunoassay (Beijing Wantai Biological Pharmacy Enterprise Co Ltd.) per manufacturer's instruction. The Wantai immunoassay, known for its high sensitivity (99%) and specificity (98%), was chosen based on its robust performance in previous HEV studies [9, 10]. Samples with standardized optical density ≥ 1.0 were interpreted as positive, < 1 as negative, and 0.9–1.1 as borderline. Borderline samples were retested, and consistently borderline samples were excluded from the analysis.

## Variable definition

The classification of province, municipality type, and ecological region adhered to Nepal's administrative guidelines. Sources of water and household toilet facilities were categorized as 'improved' or 'unimproved' based on the criteria set by the World Health Organization and the United Nations Children's Fund Joint Monitoring Programme for Water Supply, Sanitation, and Hygiene [11]. Improved sources of drinking water included piped water, boreholes or tubewells, protected dug wells, protected springs, rainwater, packaged water. Improved toilet facilities included flush/pour flush toilets connected to piped sewer systems, septic tanks or pit latrines, pit latrines with slabs (including ventilated pit latrines), and composting toilets.

The wealth quintile was derived from a previously developed abbreviated wealth index using responses to eight questions related to household assets and living conditions, such as ownership of durable goods, housing materials, and access to utilities [12]. Each item was assigned a weight based on its importance in the context of socioeconomic status, which has been calculated reference to standard wealth index in the Nepal Demographic Health Survey 2016 using 42 questions [13]. The simplified wealth index with eight questions has shown to have a kappa statistic of 0.77 to the standard wealth index [12].

## Statistical analysis

HEV seroprevalence and its 95% confidence interval were estimated based on HEV IgG seroprevalence. We estimated odds ratios (ORs) using univariate logistic regression models to explore associations between demographic factors and HEV seroprevalence. As the study was conducted using bio-banked samples from previously conducted cross-sectional survey with limited HEV risk exposure assessment, no attempt for casual inference was done. We predicted the HEV seroprevalence across Nepal in 2021 on a 5km-by-5km grid by fitting hierarchical logistic spatial regression models to observed seroprevalence data. Hierarchical logistic spatial regression models were employed to account for the multilevel structure of the data, at both the household-level and community-level. We obtained grid cell level data on population density and elevation across Nepal from the publicly available sources [14], and travel time to the nearest city from published data [15]. We used publicly available classifications of the three ecological regions [16], and subnational administrative boundaries of Nepal [17] to reflect boundaries as of November 2020. We first assigned each surveyed household to a grid cell and estimated the seroprevalence within each cell containing observations. We then fitted two models to the grid cell level seroprevalence data, firstly a model with spatial covariates for log

population density, travel time to the nearest city, and elevation, plus a spatial random effect, and secondly a null model with a spatial random effect only. The models were fitted within a Bayesian framework using integrated nested Laplace approximation (INLA) [18] and used to predict the seroprevalence in the unobserved grid cells. INLA was chosen due to its convenient propagation of uncertainty, and its computational efficiency over other Markov chain Monte Carlo based approaches to Bayesian model fitting. We used leave-one-grid-cell-out cross-validation to assess the out-of-sample predictive performance of the two models in terms of mean absolute error, bias, and the strength of the correlation between true observed seroprevalence in left out cells and the predicted seroprevalence. Seroprevalence is summarized with its 95% credible Interval (CrI). Analyses were conducted using R version 4.2.1 (The R Foundation for Statistical Computing, Vienna, Austria).

## Results

Of 3,922 participants who agreed to enroll to the original serosurvey, we tested bio-banked samples from 3,707 individuals from a representative sampling of 975 households across Nepal. The average number of household members per household was 4. After excluding four samples with consistent borderline results, final analysis was conducted with data from 3,703 participants. Detailed consort diagram for study participants is shown in Fig 1.

Of 3,707 survey participants, 3,703 remained in the final analysis after excluding those with invalid laboratory results. Overall, 20.8% (95% CI: 19.54–22.16) of the individuals tested positive for anti-HEV IgG. Seroprevalence was consistently low in children, but increased sharply with age among adults sampled. Only 5.8% (95% CI: 1.88–9.63) of males and 4.0% (95% CI: 0.16–7.76) of females were seropositive in the age group 2–5 years, which contrasts with a much higher 51.6% (95% CI: 44.4–58.9) and 42.9% (95% CI: 35.6–50.2) in males and females respectively in the age group 60+ years. Overall, male participants exhibited a higher seroprevalence of 23.5% (95% CI: 21.5–25.5) compared to females at 18.6% (95% CI: 16.9–20.3). This disparity between genders was consistent across age brackets over 20 years old. The three distinct ecological regions of Nepal exhibited variation in seroprevalence. Seroprevalence was significantly lower (P<0.001) in clusters sampled in the flat Terai region in the south of the country at 14.2% (95% CI: 12.7–15.8), compared with the Mountain region in the north of the country at 24.6% (95% CI: 18.8–30.5), and Hilly region in between at 28.9% (95% CI: 26.6–31.2). There was not a notable difference in seroprevalence between urban (20.6%, 95% CI: 19.0–22.2) and rural (21.3%, 95% CI: 19.0–23.6) municipalities (Fig 2). In univariate logistic regression analysis, we found increasing trend of OR as age increases, and male had a statistically higher risk compared to female (OR: 1.39, 95% CI: 1.17–1.65). We didn't find any statistically significant association between wealth quintile, water source, and toilet facility with HEV IgG seroprevalence (Table 1).

We fitted Bayesian geostatistical models to grid cell level seroprevalence data to create a national map of smoothed HEV seroprevalence (Fig 3). The null model, with spatial random effects only, had reasonable predictive ability with mean absolute error of 9%, a bias of 0.0034 as measured using leave one out cross-validation, and a Pearson's correlation coefficient of 0.64 between the predicted and true seroprevalence. The fully saturated model including covariates for population density, elevation, and travel time to the nearest city, did not improve predictive ability over the null model. We estimate that the percentage of people with antibodies to HEV varies widely across the country with boundaries of the three ecological regions of Nepal. An alternative geospatial map showing the boundaries of seven provinces is shown in S1 Fig. Estimates were highest in Kathmandu, reaching 50% in some grid cells, and lower than the national average in the eastern terai ecological region near the border with India at around

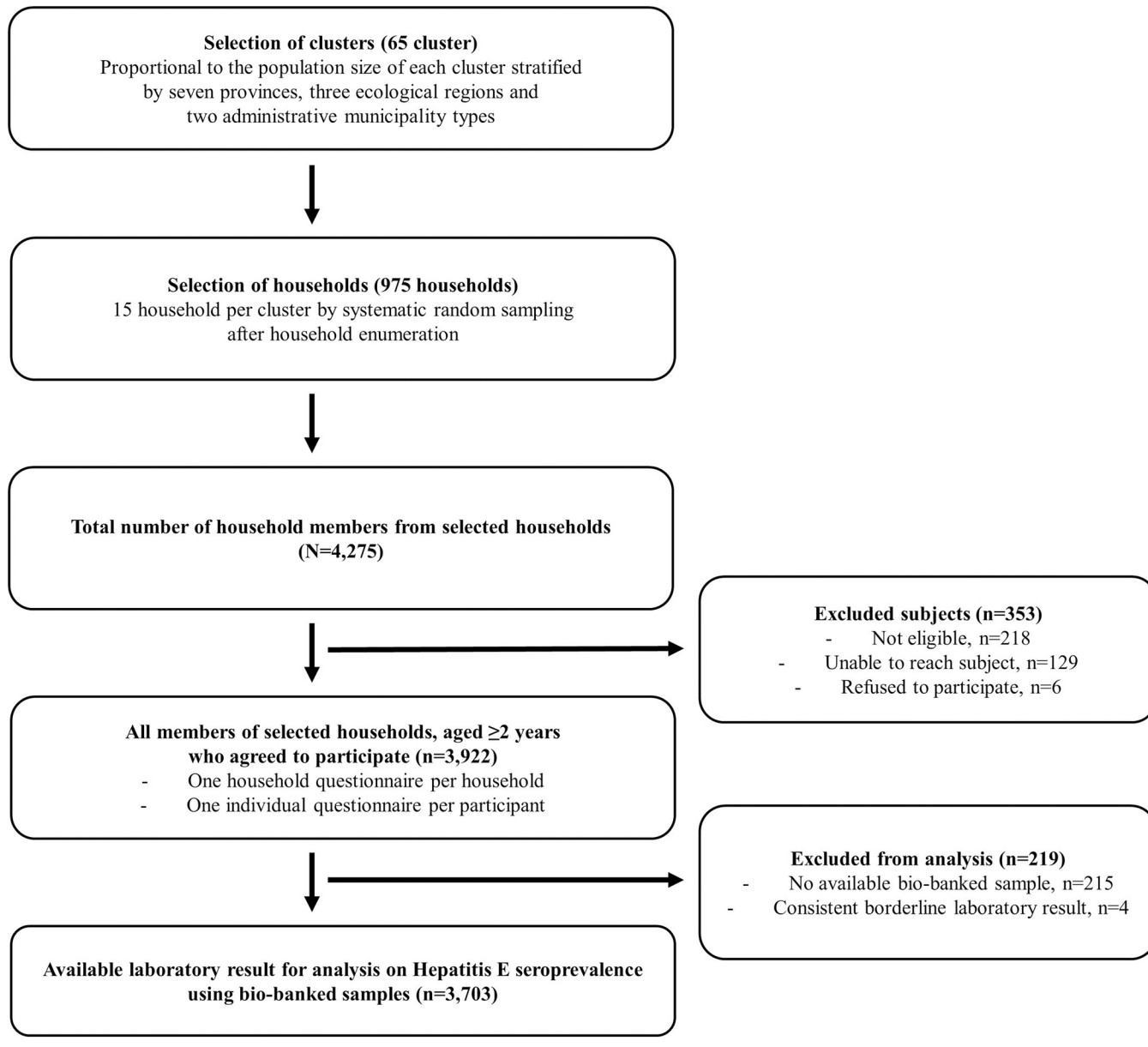

**Fig 1. Overview of Serosurvey Sampling Methodology and Consort Diagram of Study Participants.**

10%. The smooth seroprevalence estimates suggest that at least 9,732,999 people in Nepal (95% CrI: 8,827,384–10,776,403) have had prior exposure to HEV.

## Discussion

By analyzing samples from a nationally representative cross-sectional serosurvey, we found that 20.8% of the Nepali population have laboratory evidence of prior HEV infection. Based on the estimated population across Nepal and our predicted seroprevalence from our best fitting spatial model, our results suggest that almost 10 million people in Nepal have been previously infected with HEV. Although past major outbreaks have mainly been documented to affect urban areas of Nepal, we found no significant difference in seroprevalence was observed

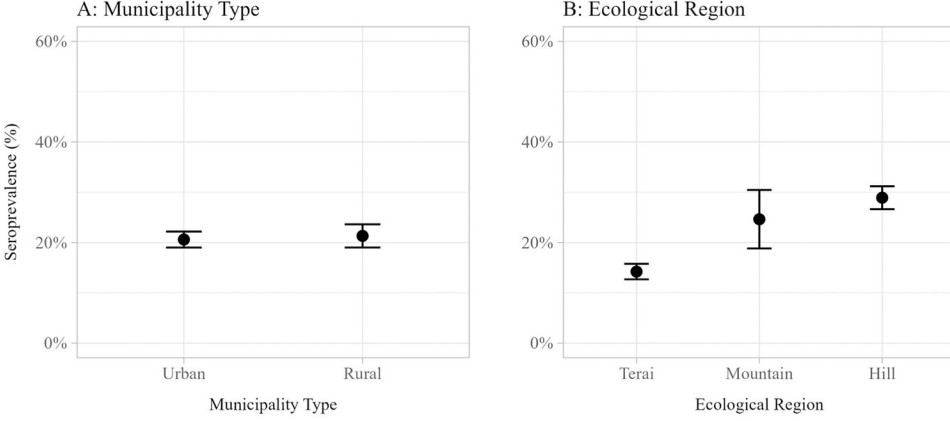

**Hepatitis E Seroprevalence by Geography**

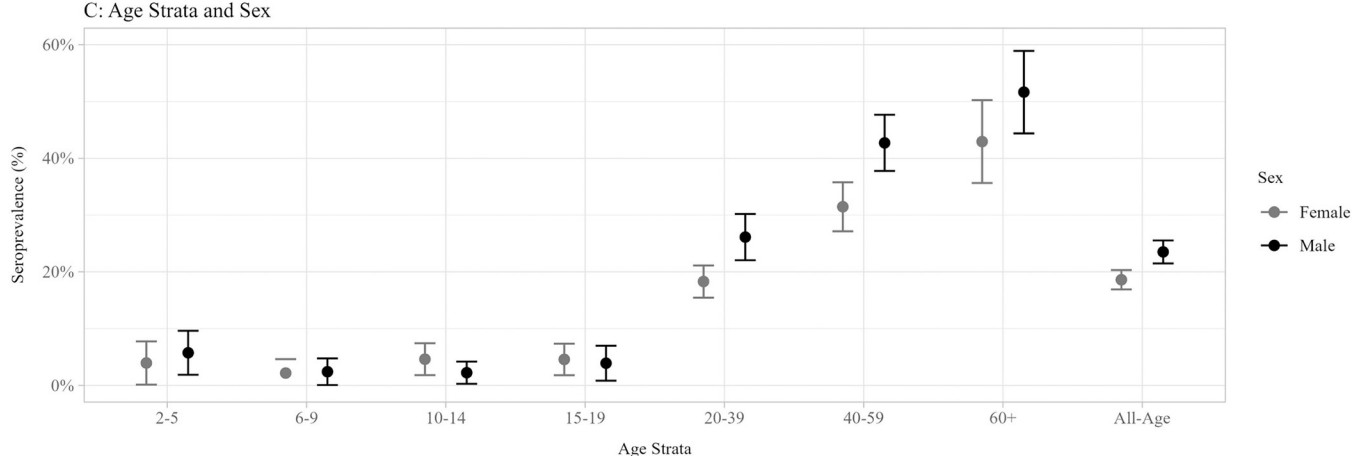

**Hepatitis E Seroprevalence by Age Strata and Sex**

**Fig 2. Hepatitis E Seroprevalence and 95% Confidence Intervals across different demographic and geographic groups.**

between urban and rural settings. However, predicted seroprevalence was estimated to be the highest in Kathmandu valley region where outbreaks have been documented previously [6, 7]. Seroprevalence varied by region and was generally higher in hilly and mountain regions as compared to terai. Lack of access to clean water may contribute to a higher HEV seroprevalence in these ecological regions as compared to terai region, as Terai is known to have higher coverage of improved water sources than mountain and hilly regions [19]. Our study findings confirmed that HEV is endemic in Nepal and suggest the need for targeted interventions in regions with high seroprevalence, particularly the hilly and mountain areas. Public health initiatives should focus on improving water quality and sanitation in these regions.

Our finding of high prevalence of HEV seroprevalence among adults is consistent with findings from the previous studies in Nepal. Overall seroprevalence of anti-HEV IgG was reported to be 54.2% from patients visiting a single tertiary hospital in Kathmandu for orthopedic injuries from January 2010 to January 2012 [20]. A study that measured HEV exposure among Nepalese blood donors reported 3.2% and 41.9% positive for HEV IgM and IgG respectively [21]. As we included children and adolescents in the analysis, we demonstrated the steep

**Table 1. Univariate Analysis of Factors Associated with Hepatitis E Virus IgG Seroprevalence.**

| Factor | Number of Participants (%) N = 3,703 | HEV IgG Seroprevalence % [CI 95%] | Crude OR [CI 95%] |
|---|---|---|---|
| **Age, y** | | | |
| 2–5 | 240 (6.48) | 5 [2.24–7.76] | — |
| 6–9 | 302 (8.16) | 2.32 [0.62–4.01] | 0.43 [0.16, 1.11] |
| 10–14 | 438 (11.83) | 3.42 [1.72–5.13] | 0.62 [0.28, 1.37] |
| 15–19 | 371 (10.02) | 4.31 [2.25–6.38] | 0.76 [0.35, 1.69] |
| 20–39 | 1,164 (31.43) | 21.31 [18.95–23.66] | 6.15 [3.30, 11.5] |
| 40–59 | 829 (22.39) | 36.67 [33.39–39.95] | 15 [7.96, 28.2] |
| 60+ | 359 (9.69) | 47.35 [42.19–52.52] | 26.9 [13.9, 52.2] |
| **Sex** | | | |
| Female | 2,010 (54.28) | 18.60 [16.90–20.30] | — |
| Male | 1,693 (45.72) | 23.50 [21.50–25.50] | 1.39 [1.17, 1.65] |
| **Province** | | | |
| Karnali | 219 (5.91) | 26.94 [21.06–32.82] | — |
| Koshi | 738 (19.93) | 13.55 [11.08–16.02] | 0.41 [0.19, 0.88] |
| Madesh | 673 (18.17) | 9.21 [7.03–11.4] | 0.26 [0.11, 0.57] |
| Bagmati* | 748 (20.2) | 37.43 [33.96–40.9] | 1.77 [0.84, 3.73] |
| Gandaki | 330 (8.91) | 15.76 [11.83–19.69] | 0.46 [0.20, 1.09] |
| Lumbini | 685 (18.5) | 20.15 [17.14–23.15] | 0.64 [0.29, 1.39] |
| Sudurpashchim | 310 (8.37) | 26.13 [21.24–31.02] | 0.94 [0.39, 2.29] |
| **Municipality** | | | |
| Urban | 2,484 (67.08) | 20.60 [19.00–22.20] | — |
| Rural | 1,219 (32.92) | 21.30 [19.00–23.60] | 1.05 [0.64, 1.73] |
| **Ecological region** | | | |
| Terai | 1,974 (53.31) | 14.20 [12.70–15.80] | — |
| Mountain | 211 (5.70) | 24.60 [18.80–30.50] | 2.32 [0.98, 5.48] |
| Hill | 1,518 (40.99) | 28.90 [26.60–31.20] | 2.8 [1.84, 4.27] |
| **Wealth quintile** | | | |
| Highest | 876 (23.66) | 25.00 [22.13–27.87] | — |
| Fourth | 1193 (32.22) | 18.52 [16.32–20.73] | 0.78 [0.60, 1.03] |
| Middle | 750 (20.25) | 16.80 [14.12–19.48] | 0.81 [0.58, 1.12] |
| Second | 446 (12.04) | 19.96 [16.25–23.66] | 0.74 [0.51, 1.08] |
| Lowest | 438 (11.83) | 26.71 [22.57–30.86] | 1.11 [0.75, 1.64] |
| **Main drinking water source** | | | |
| Improved | 3673 (99.19) | 20.77 [19.46–22.09] | — |
| Unimproved | 30 (0.81) | 30 [13.6–46.4] | 0.77 [0.31, 1.89] |
| **Toilet facility** | | | |
| Improved | 3203 (86.5) | 20.73 [19.33–22.13] | — |
| Unimproved | 500 (13.5) | 21.6 [17.99–25.21] | 0.98 [0.75, 1.27] |

HEV: Hepatitis E Virus, OR = Odds Ratio, CI = Confidence Interval

* Kathmandu is located within the Bagmati province

Water source and toilet facility are categorized as improved and unimproved according to the World Health Organization/United Nations Children's Fund classification [11].

rise in seroprevalence after age of 20 years in Nepal, suggesting infection risk may be higher in adults. This is in line with a recent longitudinal study in Kathmandu valley which measured seroincidence to be much higher in young adults than in children [22]. Low seroprevalence in

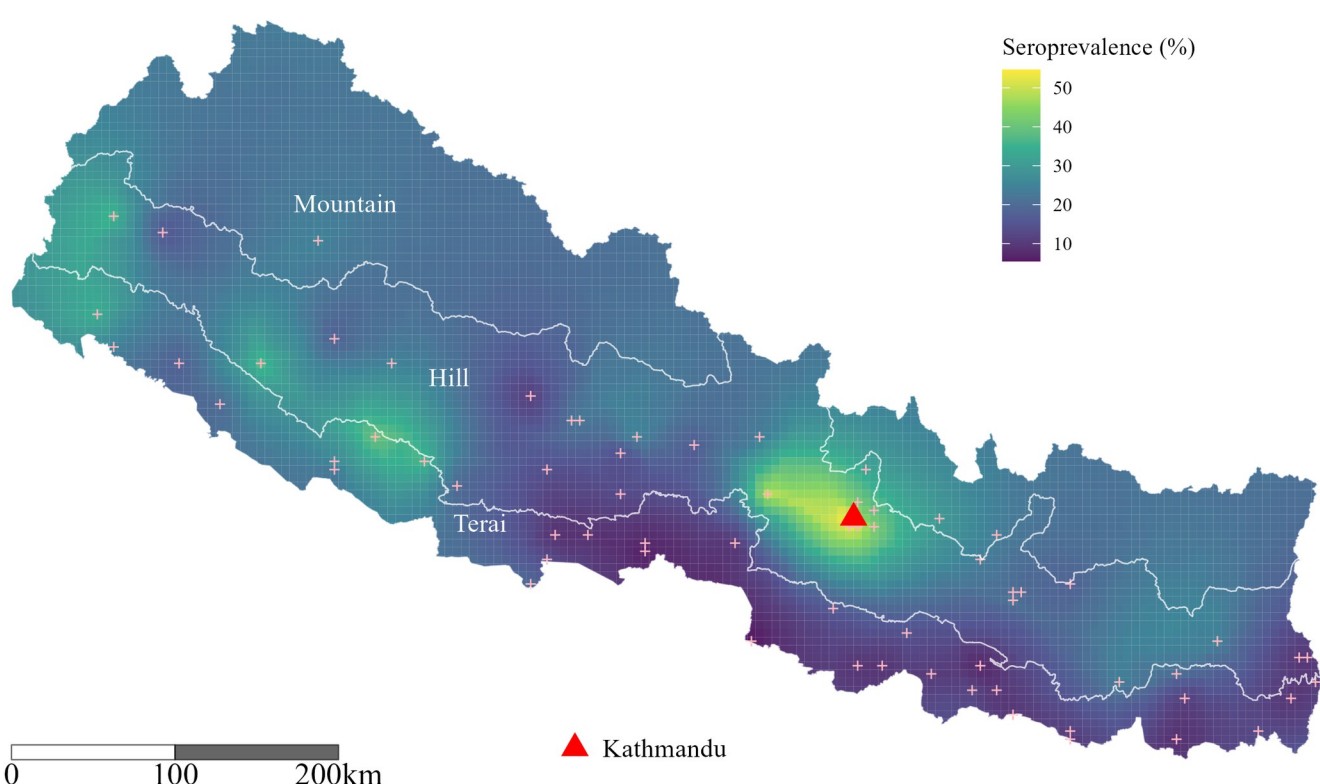

**Fig 3. Predicted seroprevalence of Hepatitis E virus across Nepal, 2021.** Predicted percentage of people with antibodies to HEV from the best fitting geostatistical model (the null model with spatial random effects only). Sampled locations are indicated by pink crosses and Kathmandu is marked with a red triangle. The map was produced in R version 4.2.1 (https://www.R-project.org/). The three ecological regions, Mountain, Hill and Terai, were defined based on the classification of the districts in the Food and Agriculture Organization's Cross sections of Nepal's physiographic regions [16]. The boundaries of the three ecological regions were made using the open-source country and district level shapefiles and spatial data made available under a Creative Commons Attribution for Intergovernmental Organisations license from the Humanitarian Data Exchange [17]. Both the open-source files and the licensing information are available at this direct link: https://data.humdata.org/dataset/cod-ab-npl.

children has also been seen in Bangladesh but seroprevalence showed a more gradual increase starting from a younger age compared to what was observed in Nepal [23].

The seroprevalence of HEV IgG as a surrogate marker for past infection may underestimate the true infection rate, as antibodies against HEV are known to wane over time. This decline in antibody levels, known as seroreversion, could lead to false negatives, particularly among those infected in the distant past. Long-term HEV vaccine efficacy study in China found that although vaccine-induced antibodies persisted for at least 8.5 years, for seropositive individuals in the placebo group the geometric mean antibody concentration decreased to less than half the baseline over the same period [24]. Another study in Bangladesh reported antibody loss, i.e., seroreversion, of 20% 10 years post HEV infection, and the seroreversion was higher among those who were infected at younger age [25]. A recent study that aimed to measure the rate of HEV seroreversion in Bangladesh reported the annual rate to be 15% (95% CI: 10–21%) [26]. This same study found that children are more likely to serorevert which could contribute to the lower HEV IgG seroprevalence among young children in our data.

In endemic areas, both large outbreaks and sporadic cases of HEV infections occur frequently usually due to genotype 1 or 2 HEV via fecal to oral transmission through contamination of drinking water or contaminated food [27]. The only licensed vaccine against HEV is Hecolin (Xiamen Innovax Biotech) for people 16–65 years of age which demonstrated 100%

(95% CI 72.1–100.0) efficacy in a Phase 3 study conducted in China [28]. Currently this vaccine is only registered in China and Pakistan. The World Health Organization recommends consideration of the use of HEV vaccine where risk of complications or mortality is high to mitigate or prevent HEV outbreaks especially in high-risk groups such as pregnant women [29]. Although those >20 years have the highest seroprevalence, there may still a substantial burden of infection apparent in younger children. The recent findings of HEV IgG seroreversion rate and relatively shorter longevity of antibodies among children further suggest seroprevalence studies may underrepresent their history of infection. Children should be further studied to identify the incidence of infection and the role they play in transmission. This data would have implications for the need to de-escalate the age indication for the vaccine if symptomatic or asymptomatic children play a role in HEV transmission dynamics.

Our study is subject to several limitations. Even though we deployed representative sampling strategy, we noticed some differences to the 2021 national census data [30]. While the proportion of study participants by age strata, ecological region was similar to that of the national census, our sampled population slightly over-represented female and under-represented children. As much as we aimed to collect blood sample from every household member by accommodating the most convenient time for every eligible household member, there was a greater challenge to reach male participants esp. those who spend majority of days in other regions for work-related reasons. We also observed higher study refusal rate among children. In addition, household-based sampling may systematically exclude migrant populations and those living in informal settlements who may be at higher risk for HEV infection due to poor access to water and sanitation facilities. As our study used bio-banked samples from other study, no specific questions on the history of jaundice or acute hepatitis were included, limiting our ability to assess HEV associated factors. In addition, as there is no commercially available laboratory assay that can differentiate between HEV genotypes serologically, therefore we were unable to determine which genotype or genotypes caused past infection in seropositive participants. Sequenced isolates from past outbreaks and sporadic cases in Nepal have all been classified as genotype 1 [8, 31–33], but HEV has previously been detected in domesticated pigs in the Kathmandu valley [34] and genotype 4 is known to circulate in swine in neighboring northern India [35,36]. It remains unclear to what extent zoonotic transmission is occurring in Nepal, and future study on HEV in Nepal should aim to classify HEV infections by genotypes. Lastly, the study presented estimates of the HEV seroprevalence throughout the country based on data from only 65 sampled wards using a geostatistical model which assumes that seroprevalence changes smoothly across space. Although we were able to account for elevation, population density and travel time to the nearest city in the fully saturated model, in reality, there are likely to be more spatial factors for which data was not available that may mean the seroprevalence does not change smoothly.

## Conclusion

The results of this nationally representative serosurvey for HEV in Nepal provide population-based serologic evidence that HEV is endemic in the country, with the highest risk of infection within Kathmandu valley. The considerable geographic variation that we observed in the measured seroprevalence underscores the need for targeted surveillance and intervention programs to make efficient use of limited resources. Targeted strengthening water, sanitation, and hygiene (WASH) initiatives along with potential use of HEV vaccine should be considered in areas with high seroprevalence for high-risk groups, or during an acute outbreak situation, to reduce the burden of HEV in Nepal.

## Supporting information

**S1 Fig. Predicted seroprevalence of Hepatitis E virus across Nepal, 2021.** Predicted percentage of people with antibodies to HEV from the best fitting geostatistical model (the null model with spatial random effects only). Sampled locations are indicated by pink crosses and Kathmandu is marked with a red triangle. The map was produced in R version 4.2.1 (https://www.R-project.org/). The boundaries of the seven provinces of Nepal were made using the open-source shapefiles and spatial data made available under a Creative Commons Attribution for Intergovernmental Organisations license from the Humanitarian Data Exchange [17]. Both the open-source files and the licensing information are available at this direct link: https://data.humdata.org/dataset/cod-ab-npl.
(TIF)

## Acknowledgments

We thank all the participants in the cross-sectional survey for taking part in this research. We thank the field team at New Era and laboratory staff at Nepal Medical College and Dhulikhel Hospital for their dedicated work.

## Author Contributions

**Conceptualization:** Chulwoo Rhee, Amy Dighe, Andrew S. Azman, Julia Lynch.

**Data curation:** Chulwoo Rhee, Amy Dighe, Nishan Katuwal, Dipesh Tamrakar, Jacqueline KyungAh Lim, Nimesh Poudyal, Ritu Amatya, Rajeev Shrestha, Andrew S. Azman, Julia Lynch.

**Formal analysis:** Chulwoo Rhee, Amy Dighe, Haeun Cho, Ramzi Mraidi, Il-Yeon Park, Deok Ryun Kim, Andrew S. Azman, Julia Lynch.

**Funding acquisition:** Chulwoo Rhee, Julia Lynch.

**Investigation:** Chulwoo Rhee, Jacqueline KyungAh Lim, Nimesh Poudyal, Ritu Amatya.

**Methodology:** Chulwoo Rhee, Amy Dighe, Andrew S. Azman, Julia Lynch.

**Supervision:** Chulwoo Rhee, Nishan Katuwal, Dipesh Tamrakar, Jacqueline KyungAh Lim, Nimesh Poudyal, Ritu Amatya, Rajeev Shrestha.

**Validation:** Nishan Katuwal, Dipesh Tamrakar, Rajeev Shrestha.

**Visualization:** Chulwoo Rhee, Amy Dighe, Haeun Cho, Ramzi Mraidi, Il-Yeon Park, Deok Ryun Kim.

**Writing – original draft:** Chulwoo Rhee, Amy Dighe.

**Writing – review & editing:** Chulwoo Rhee, Amy Dighe, Nishan Katuwal, Haeun Cho, Ramzi Mraidi, Dipesh Tamrakar, Jacqueline KyungAh Lim, Nimesh Poudyal, Il-Yeon Park, Deok Ryun Kim, Ritu Amatya, Rajeev Shrestha, Andrew S. Azman, Julia Lynch.

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
