## [Decision Letter · Decision Letter 0]

2 Aug 2024

Dear Mr Rhee,

Thank you very much for submitting your manuscript "Seroprevalence and Geospatial Distribution of Hepatitis E Seropositivity in Nepal, 2021" for consideration at PLOS Neglected Tropical Diseases. As with all papers reviewed by the journal, your manuscript was reviewed by members of the editorial board and by several independent reviewers. In light of the reviews (below this email), we would like to invite the resubmission of a significantly-revised version that takes into account the reviewers' comments. 

Please address all the reviewers' comments point by point, either by incorporating their suggestions into the manuscript, or by giving an explanation of why you disagree with their assessment. You do not need to make any substantial changes to the analysis unless you feel that it would both address a reviewer's comment and improve the quality of the paper.

We cannot make any decision about publication until we have seen the revised manuscript and your response to the reviewers' comments. Your revised manuscript is also likely to be sent to reviewers for further evaluation.

Sincerely,

Mabel Carabali, M.D., M.Sc., Ph.D.,

Section Editor

Mabel Carabali

Section Editor

Please address all the reviewers' comments point by point, either by incorporating their suggestions into the manuscript, or by giving an explanation of why you disagree with their assessment. You do not need to make any substantial changes to the analysis unless you feel that it would both address a reviewer's comment and improve the quality of the paper.

Reviewer's Responses to Questions

**Key Review Criteria Required for Acceptance?**

**Methods**

-Are the objectives of the study clearly articulated with a clear testable hypothesis stated?

-Is the study design appropriate to address the stated objectives?

-Is the population clearly described and appropriate for the hypothesis being tested?

-Is the sample size sufficient to ensure adequate power to address the hypothesis being tested?

-Were correct statistical analysis used to support conclusions?

-Are there concerns about ethical or regulatory requirements being met?

Reviewer #1: -Are the objectives of the study clearly articulated with a clear testable hypothesis stated?

Yes

-Is the study design appropriate to address the stated objectives?

Yes 

-Is the population clearly described and appropriate for the hypothesis being tested?

Yes

-Is the sample size sufficient to ensure adequate power to address the hypothesis being tested?

Yes

-Were correct statistical analysis used to support conclusions?

Yes 

-Are there concerns about ethical or regulatory requirements being met?

Yes

Reviewer #2: Is there any information about travel history from the study participants?

Reviewer #3: Methods

Survey Design:

• Lines 85-103: Clarify the criteria for selecting wards and households more explicitly. Explain if there were any specific characteristics considered in the selection process beyond population size.

– Suggested addition: “Wards were selected based on population size and geographic diversity to ensure representativeness across different ecological regions and municipality types. Households within these wards were then chosen using systematic random sampling, taking into account factors such as household density and accessibility.”

Laboratory Analysis:

• Lines 104-110: Include details about the sensitivity and specificity of the Wantai immunoassay used for testing anti-HEV IgG. Explain why this particular assay was chosen over others.

– Suggested addition: “The Wantai immunoassay, known for its high sensitivity (99%) and specificity (98%), was chosen based on its robust performance in previous HEV studies.”

• Lines 107-109: Clarify the handling of borderline samples.

– Current: “Borderline samples were retested, and consistently borderline samples were excluded from the analysis.”

– Suggested: “Samples with optical density values close to the cutoff (0.9-1.1) were retested, and those consistently remaining borderline were excluded from the final analysis to ensure accuracy.”

Variable Definition:

• Lines 112-123: Elaborate on the wealth index calculation for better reproducibility. Include a brief description of the eight questions used and how responses were weighted.

– Suggested addition: “The wealth index was derived from responses to eight questions related to household assets and living conditions, such as ownership of durable goods, housing materials, and access to utilities. Each item was assigned a weight based on its importance in the context of socioeconomic status, following the methodology of the Nepal Demographic Health Survey 2016.”

• Lines 115-118: Rephrase for clarity.

– Current: “Sources of water and household toilet facility were categorized as improved and unimproved according to the World Health Organization/United Nations Children’s Fund Joint Monitoring Programme for Water Supply, Sanitation and Hygiene criteria.”

– Suggested: “Sources of water and household toilet facilities were categorized as ‘improved’ or ‘unimproved’ based on the criteria set by the World Health Organization and the United Nations Children’s Fund Joint Monitoring Programme for Water Supply, Sanitation, and Hygiene.”

**Results**

-Does the analysis presented match the analysis plan?

-Are the results clearly and completely presented?

-Are the figures (Tables, Images) of sufficient quality for clarity?

Reviewer #1: -Does the analysis presented match the analysis plan?

Yes 

-Are the results clearly and completely presented?

Yes

-Are the figures (Tables, Images) of sufficient quality for clarity?

Yes

Reviewer #2: 1. I suggest to incorporate the population density of the different sampling sites as a factor for the HEV seroprevalence study and to include this in the discussion. This could be incorporated in Table 2 and as a graph.

2. The provinces mentioned in Table 2 are from Kathmandu? Please describe this in the text.

3. An additional figure could be included showing the seroprevalence across the 7 provinces studied.

Reviewer #3: Statistical Analysis:

• Lines 125-143: Provide a justification for using hierarchical logistic spatial regression models and discuss any assumptions made. Explain why Bayesian models were chosen and the advantages they offer in this context.

– Suggested addition: “Hierarchical logistic spatial regression models were employed to account for the multilevel structure of the data, capturing both individual-level and area-level variations in HEV seroprevalence. Bayesian methods were chosen for their ability to incorporate prior information and provide robust estimates even with complex spatial dependencies. This approach also allows for more accurate uncertainty quantification in the presence of spatial autocorrelation.”

Results

• Lines 145-177: Highlight only key statistical results (e.g., p-values) in the text for important comparisons. Explicitly state the statistical significance of differences in seroprevalence between regions.

– Suggested addition: “The differences in seroprevalence between the hilly (28.9%, 95% CI: 26.6-31.2) and terai (14.2%, 95% CI: 12.7-15.8) regions were statistically significant (p < 0.001), indicating a higher burden of HEV in the hilly regions.”

Tables and Figures:

• Ensure all figures are high resolution and clearly labeled. Consider adding more descriptive captions that summarize the key takeaways from each figure/table.

– Suggested addition for Table 1 caption: “Table 1. Demographic characteristics of study participants and HEV seroprevalence by demographic factors.”

– Suggested addition for Figure 1 caption: “Figure 1. Hepatitis E Seroprevalence and 95% Confidence Intervals across different demographic and geographic groups.”

**Conclusions**

-Are the conclusions supported by the data presented?

-Are the limitations of analysis clearly described?

-Do the authors discuss how these data can be helpful to advance our understanding of the topic under study?

-Is public health relevance addressed?

Reviewer #1: -Are the conclusions supported by the data presented?

Yes

-Are the limitations of analysis clearly described?

Yes 

-Do the authors discuss how these data can be helpful to advance our understanding of the topic under study?

Yes

-Is public health relevance addressed?

Yes

Reviewer #2: It is not clear how authors estimate that around 10 million people in Nepal have had previous exposure to HEV.

Reviewer #3: Conclusion

• Lines 249-255: Reinforce the importance of the study’s findings for public health planning and resource allocation. Suggest specific areas for intervention based on the geographic distribution of HEV seropositivity.

– Suggested addition: “The high seroprevalence in Kathmandu and other urban areas underscores the urgent need for targeted surveillance and intervention programs. Strengthening water, sanitation, and hygiene (WASH) initiatives and considering HEV vaccination for high-risk groups are crucial steps for reducing the burden of HEV in Nepal.”

**Editorial and Data Presentation Modifications?**

Reviewer #1: The paper by Rhee and coauthors presents the results of a nationally representative cross-sectional hepatitis E serosurvey in Nepal. The study is well researched and provides the comprehensive analysis of hepatitis E virus seroprevalence in the country. The manuscript is written well. I only have some minor comments:

1. L.20. Please indicate that this is IgG antibodies that are indicative of past infection.

2. L.23 – Please indicate in the abstract the real number of serum samples used in the HEV seroprevalence study (3,703)

3. L.42 and L.58. Person-to-person transmission is usually understood as a direct transmission via close contacts between persons and is rare in hepatitis E. Therefore, this term should be avoided in description of waterborne transmission. Perhaps, authors mean that humans are the source of this infection contrary to zoonotic HEV genotypes. 

4. Please indicate if the bio-banked samples were stored at aliquots, at what temperature and how many times samples were thawed/frozen before the anti-HEV antibody testing.

5. L.108. Was it optical denstity or OD sample/cut-off OD value?

Reviewer #2: (No Response)

Reviewer #3: (No Response)

**Summary and General Comments**

Reviewer #1: The study described in the manuscript provides the comprehensive analysis of hepatitis E virus seroprevalence in Nepal. The data obtained in the study confirmed that hepatitis E is endemic in the country, especially in Katmandu region, due to poor sanitation and limited access to quality drinking water. This situation indicate the need for licensed HEV vaccines in the country. In this regard, the study has a significance for public health.

Reviewer #2: The manuscript "Seroprevalence and Geospatial Distribution of Hepatitis E Seropositivity in Nepal, 2021" reports a wide serological surveillance for HEV in Nepal which can contribute to expand the HEV epidemiological data in this country. 

It is generally well presented, however there are some issues that need to be considered in order to improve the work.

Reviewer #3: Overall Assessment

The manuscript is well-structured and provides significant insights into HEV seroprevalence in Nepal. Addressing the suggestions mentioned above will further enhance the manuscript’s clarity, context, and impact. The detailed proofreading notes aim to improve readability and precision.

Recommendations

1. Minor Revision: Address the suggestions and proofreading notes provided above.

2. Ethical Approval: Ensure all ethical considerations are clearly documented, particularly regarding bio-banking and informed consent.

3. Future Research: Include a brief discussion on future research directions and policy implications based on the findings. Provide more detailed recommendations for policymakers and public health practitioners.

Specific Areas for Improvement

1. Introduction: Provide more background on HEV epidemiology in Nepal and previous studies to set the stage for the current research.

2. Methods: Clarify sampling criteria, provide more detail on laboratory quality control, and justify the choice of statistical models.

3. Results: Highlight key statistical findings more clearly and discuss observed patterns in greater detail.

4. Discussion: Expand on policy implications and future research directions. Address limitations in more depth.

5. Figures and Tables: Ensure high-resolution images and add more descriptive captions. Consider including additional visualizations to illustrate regional differences.

6. References: Update your references to the most recent literature, the oldest shoud be five years ago and not beyond that, also ensure consistency in formatting. Include more recent studies.

PLOS authors have the option to publish the peer review history of their article (what does this mean?). If published, this will include your full peer review and any attached files.

Reviewer #1: No

Reviewer #2: No

Reviewer #3: No
---

## [Editor Report · Decision Letter 1]

8 Nov 2024

Dear Dr Rhee and colleagues,

Thank you for addressing many of the comments from the review of your article. Before we reach a decision, please could you clarify your response to the following comment from reviewer 1: "Please indicate if the bio-banked samples were stored at aliquots, at what temperature and how many times samples were thawed/frozen before the anti-HEV antibody testing."

It was not clear to me how this was addressed. If you are unable to provide the information or think that it is not relevant, please explain why. If the information is already available from another source (e.g. the protocol mentioned), please direct the reader to it with appropriate citations (e.g. procedures for storing and handling the bio-banked samples have previously been published in...")

Once we have your response to this, we will proceed to a decision.

Many thanks,

Josh M Colston, Ph.D.

Academic Editor

---

## [Editor Report · Decision Letter 2]

30 Nov 2024

Dear Mr Rhee,

We are pleased to inform you that your manuscript 'Geospatial Distribution of Hepatitis E Seroprevalence in Nepal, 2021' has been provisionally accepted for publication in PLOS Neglected Tropical Diseases.

Best regards,

Josh M Colston, Ph.D.

Academic Editor

Mabel Carabali

Section Editor

Shaden Kamhawi

co-Editor-in-Chief

Paul Brindley

co-Editor-in-Chief

Thank you for addressing the remaining comments. Having judged that the reviews have been adequately addressed, we have decided to accept your article for publication in PLOS NTDs. Congratulations.

---

## [Editor Report · Acceptance letter]

13 Dec 2024

Dear Mr Rhee,

We are delighted to inform you that your manuscript, "Geospatial Distribution of Hepatitis E Seroprevalence in Nepal, 2021," has been formally accepted for publication in PLOS Neglected Tropical Diseases.

Best regards,

Shaden Kamhawi

co-Editor-in-Chief

Paul Brindley

co-Editor-in-Chief
